# MindEye2: Shared-Subject Models Enable FMRI-To-Image With 1 Hour of Data

Paul S. Scotti[1,2], Mihir Tripathy [*2], Cesar Kadir Torrico Villanueva [*2], Reese Kneeland [*2,4], Tong Chen[2,5], Ashutosh Narang[2], Charan Santhirasegaran[2], Jonathan Xu[2,6], Thomas Naselaris[4], Kenneth A. Norman[3], and Tanishq Mathew Abraham[1,2]

[1]Stability AI
[2]Medical AI Research Center (MedARC)
[3]Princeton Neuroscience Institute
[4]University of Minnesota
[5]The University of Sydney
[6]University of Waterloo

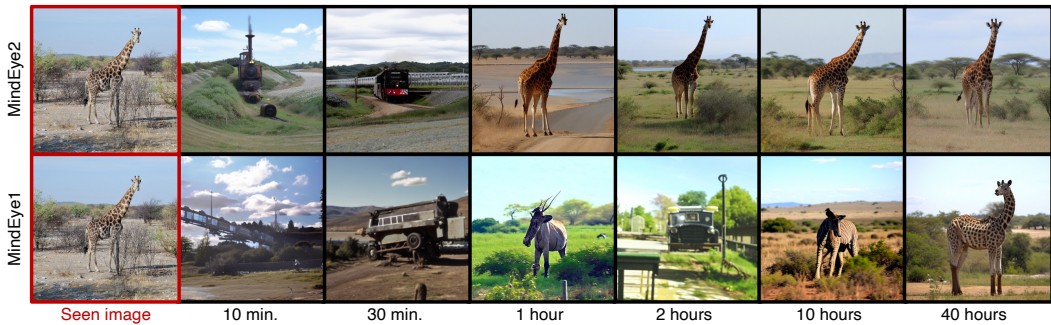

Figure 1: MindEye2 vs. MindEye1 reconstructions from fMRI using varying amounts of subject 1 data.

## Abstract

Reconstructions of visual perception from brain activity have improved tremendously, but the practical utility of such methods has been limited. This is because such models are trained independently per subject where each subject requires dozens of hours of expensive fMRI training data to attain high-quality results. The present work showcases high-quality reconstructions using only 1 hour of fMRI training data. We pretrain our model across 7 subjects and then fine-tune on minimal data from a new subject. Our novel functional alignment procedure linearly maps all brain data to a shared-subject latent space, followed by a shared non-linear mapping to CLIP image space. We then map from CLIP space to pixel space by fine-tuning Stable Diffusion XL to accept CLIP latents as inputs instead of text. This approach improves out-of-subject generalization with limited training data and also attains state-of-the-art image retrieval and reconstruction metrics compared to single-subject approaches. MindEye2 demonstrates how accurate reconstructions of perception are possible from a single visit to the MRI facility. All code is available on OSF.

## 1 Introduction

Spurred by the open releases of deep learning models such as CLIP Radford et al. (2021) and Stable Diffusion Rombach et al. (2022), along with large-scale functional magnetic resonance imaging (fMRI) datasets such as the Natural Scenes Dataset Allen et al. (2022) where human participants were scanned viewing tens of thousands of images, there has been an influx of research papers

---

*Core contribution.

demonstrating the ability to reconstruct visual perception from brain activity with high fidelity Scotti et al. (2023); Ozcelik and VanRullen (2023); Ozcelik et al. (2022); Takagi and Nishimoto (2022; 2023); Gu et al. (2023); Kneeland et al. (2023a;b;c); Ferrante et al. (2023a); Thual et al. (2023); Chen et al. (2023a;b); Sun et al. (2023); Gaziv et al. (2022); Mai and Zhang (2023); Xia et al. (2023). FMRI indirectly measures neural activity by detecting changes in blood oxygenation. These patterns of fMRI brain activity are translated into embeddings of pretrained deep learning models and used to visualize internal mental representations Beliy et al. (2019); Shen et al. (2019a;b); Seeliger et al. (2018); Lin et al. (2019).

Visualization of internal mental representations, and more generally the ability to map patterns of brain activity to the latent space of rich pretrained deep learning models, has potential to enable novel clinical assessment approaches and brain-computer interface applications. However, despite all the recent research demonstrating high-fidelity reconstructions of perception, the practical adoption of such approaches to these settings has been limited if not entirely absent. A major reason for this is that the high-quality results shown in these papers use single-subject models that are not generalizable across people, and which have only been shown to work well if each subject contributes dozens of hours of expensive fMRI training data. MindEye2 introduces a novel functional alignment procedure that overcomes these barriers by pretraining a shared-subject model that can be fine-tuned using limited data from a held-out subject and generalizing to held-out data from that subject. This approach yields similar reconstruction quality to a single-subject model trained using $40\times$ the training data. See Figure 1 for selected samples of reconstructions obtained from just 1 hour of data from subject 1 compared to their full 40 hours of training data in the Natural Scenes Dataset.

In addition to a novel approach to shared-subject alignment, MindEye2 builds upon the previous SOTA approach introduced by MindEye1 Scotti et al. (2023). In terms of similarities, both approaches map flattened spatial patterns of fMRI activity across voxels (3-dimensional cubes of cortical tissue) to the image embedding latent space of a pretrained CLIP Radford et al. (2021) model with the help of a residual MLP backbone, diffusion prior, and retrieval submodule. The diffusion prior Ramesh et al. (2022) is used for reconstruction and is trained from scratch to take in the outputs from the MLP backbone and produce aligned embeddings suitable as inputs to any pretrained image generation model that accepts CLIP image embeddings (hereafter referred to as unCLIP models). The retrieval submodule is contrastively trained and produces disjointed CLIP-fMRI embeddings that can be used to find the original (or nearest neighbor) image in a pool of images, but is not used to reconstruct a novel image. Both MindEye2 and MindEye1 also map brain activity to the latent space of Stable Diffusion's Rombach et al. (2022) variational autoencoder (VAE) to obtain blurry reconstructions that lack high-level semantic content but perform well on low-level image metrics (e.g., color, texture, spatial position), which get combined with the semantically rich outputs from the diffusion prior to return reconstructions that perform well across perceptual and semantic features.

MindEye2 innovates upon MindEye1 in the following ways: (1) Rather than the whole pipeline being independently trained per subject, MindEye2 is pretrained on data from other subjects and then fine-tuned on the held-out target subject. (2) We map from fMRI activity to a richer CLIP space provided by OpenCLIP ViT-bigG/14 Schuhmann et al. (2022); Ilharco et al. (2021), and reconstruct images via a fine-tuned Stable Diffusion XL unCLIP model that supports inputs from this latent space. (3) We merge the previously independent high- and low-level pipelines into a single pipeline through the use of submodules. (4) We additionally predict the text captions of images to be used as conditional guidance during a final image reconstruction refinement step.

The above changes support the following main contributions of this work: (1) Using the full fMRI training data from Natural Scenes Dataset we achieve state-of-the-art performance across image retrieval and reconstruction metrics. (2) Our novel multi-subject alignment procedure enables competitive decoding performance even with only 2.5% of a subject's full dataset (i.e., 1 hour of scanning).

## 2 MINDEYE2

MindEye2 involves pretraining and then fine-tuning a single model where brain activity is mapped to the embeddings of pretrained deep learning models. During inference, these embeddings predicted from the brain are fed into frozen image generative models that translate from model space to pixel space. Our strategy to reconstruct seen images from brain activity using minimal training data is to

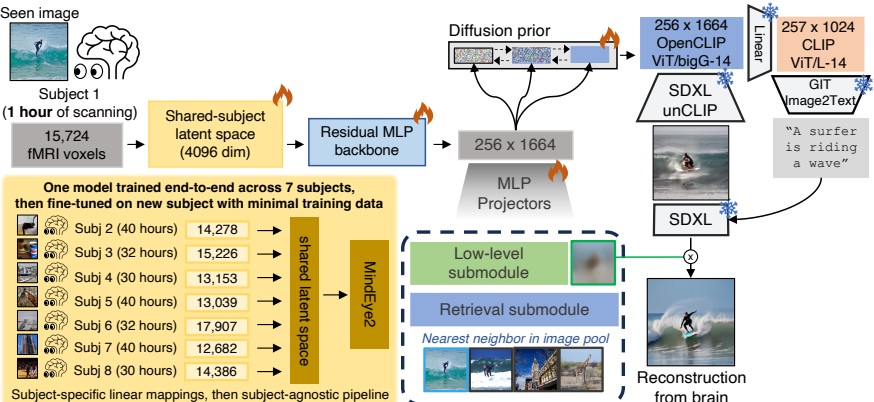

Figure 2: MindEye2 overall schematic. MindEye2 is trained using samples from 7 subjects in the Natural Scenes Dataset and then fine-tuned using a target held-out subject who may have scarce training data. Ridge regression maps fMRI activity to an initial shared-subject latent space. An MLP backbone and diffusion prior output OpenCLIP ViT-bigG/14 embeddings which SDXL unCLIP uses to reconstruct the seen image, which are then refined with base SDXL. The submodules help retain low-level information and support retrieval tasks. Snowflakes=frozen models used during inference, flames=actively trained.

first pretrain the model using data from 7 subjects (30-40 hours of scanning data each) and then to fine-tune the model using data from a held-out 8th subject. The full MindEye2 pipeline is depicted in Figure 2.

Single-subject models were trained/fine-tuned on a single 8xA100 80Gb GPU node for 150 epochs with a batch size of 24. Multi-subject pretraining was done with a batch size of 63 (9 samples per each of 7 subjects). Models were trained with Huggingface Accelerate Gugger et al. (2022) and DeepSpeed Rajbhandari et al. (2020) Stage 2 with CPU offloading.

## 2.1 Shared-Subject Functional Alignment

Every subject has a uniquely shaped brain with different functional organization, meaning that there needs to be an initial alignment step to ensure the model can handle inputs from different brains. Unlike anatomical alignment where every subject's brain is mapped to the same brain template Talairach and Tournoux (1990); Mazziotta et al. (2001), we remain in subjects' native brain space and functionally align flattened spatial patterns of fMRI activity to a shared-subject latent space using subject-specific ridge regression. That is, each subject has a separate linear layer with weight decay to map the input fMRI voxels ($13,000$ to $18,000$ voxels depending on the subject) to a 4096-dim latent.

Following this initial linear layer, the rest of the model pipeline is shared across subjects without any subject-specific mappings. The alignment to shared-subject space is not trained independently—the whole pipeline is trained end-to-end where pretraining involves each batch containing brain inputs from all subjects. That is, we do not pretrain a model for each subject; rather, we pretrain a single model equally sampling across all the subjects except the held-out subject used for fine-tuning.

Two strengths of this novel functional alignment procedure are in its simplicity and flexibility. Using a simple linear mapping for alignment can provide robust, generalizeable performance in low-sample, high-noise settings because simple mappings are less likely to overfit to noise. Also, unlike typical functional alignment approaches that require subjects to process a shared set of images Haxby et al. (2011), our approach has the flexibility to work even when subjects are viewing entirely unique images in the training data. This is critical for the Natural Scenes Dataset, where 90% of the seen images are unique to the subject and the 10% that were seen across subjects are relegated to the test set. Further, this approach holds advantages for subsequent data collection of a new subject, where such data collection does not need to be restricted to showing a predefined set of images.

## 2.2 BACKBONE, DIFFUSION PRIOR, & SUBMODULES

Aside from linear mapping to a shared-subject space, mapping to OpenCLIP ViT-bigG/14 rather than CLIP ViT-L/14, and adding a low-level MLP submodule, the MindEye2 model is trained identically to MindEye1. Flattened spatial patterns of brain activity are first linearly mapped to the shared-subject space using an output dimensionality of 4096. Then, these latents are fed through an MLP backbone with 4 residual blocks, followed by a linear mapping that goes from 4096-dim to $256 \times 1664$ dimensionality of OpenCLIP ViT-bigG/14 image token embeddings. These backbone embeddings are then simultaneously fed through a diffusion prior Ramesh et al. (2022) and two MLP projectors (retrieval and low-level submodules).

MindEye2 has three losses that are summed, stemming from the diffusion prior, retrieval submodule, and low-level submodule. The end-to-end loss, with $\alpha_1 = .033$ and $\alpha_2 = .016$, is defined as:

$$\mathcal{L} = \mathcal{L}_{\text{prior}} + \alpha_1 \cdot \mathcal{L}_{\text{BiMixCo|SoftCLIP}} + \alpha_2 \cdot \mathcal{L}_{\text{lowlevel}} \tag{1}$$

### 2.2.1 DIFFUSION PRIOR

Using a diffusion prior to align outputs from a contrastive learning model was inspired by DALL-E 2 Ramesh et al. (2022), where a "diffusion prior" maps CLIP text embeddings to CLIP image space before using an unCLIP decoder to reconstruct images. Here we trained our own diffusion prior from scratch to map fMRI latents to the OpenCLIP ViT-bigG/14 image space, which was kept frozen as done with locked-image text tuning (LiT) Zhai et al. (2022). We used the same prior loss as Ramesh et al. (2022), implemented with the same code as MindEye1 which used modified code from the DALLE2-pytorch repository.

### 2.2.2 RETRIEVAL SUBMODULE

MindEye1 observed a tradeoff if using contrastive loss and MSE loss on the outputs of the diffusion prior directly, such that the model could not effectively learn a single embedding to satisfy both objectives. Instead, applying MSE loss on the diffusion prior and applying contrastive loss on the outputs from an MLP projector attached to the MLP backbone effectively mitigated this tradeoff because the objectives no longer shared identical embeddings. We adopted the same approach here, with the retrieval submodule contrastively trained to maximize cosine similarity for positive pairs while minimizing similarity for negative pairs. We used the same BiMixCo and SoftCLIP losses used in MindEye1 Scotti et al. (2023), which involved the first third of training iterations using bidirectional MixCo data augmentation Kim et al. (2020) and the last two-thirds of training iterations using soft labels (generated from the dot product of CLIP image embeddings in a batch with themselves) without data augmentation.

### 2.2.3 LOW-LEVEL SUBMODULE

MindEye1 used an independent low-level pipeline to map voxels to the latent space of Stable Diffusion's variational autoencoder (VAE) such that blurry reconstructions were returned that lacked semantic information but performed well on low-level metrics. Here, we reimplement this pipeline as a submodule, similar to the retrieval submodule, such that it need not be trained independently. The MLP projector feeds to a CNN upsampler that upsamples to the $(64, 64, 4)$ dimensionality of SD VAE latents with L1 loss as well as an additional MLP to the embeddings of a teacher linear segmentation model VICRegL Bardes et al. (2022) ConvNext-XXL ($\alpha = 0.75$) for an auxilliary SoftCLIP loss (soft labels from VICRegL model).

$$\mathcal{L}_{\text{lowlevel}} = \frac{1}{N} \sum_{i=1}^{N} |\text{VAE}_i - \hat{\text{VAE}}_i| + L_{\text{SoftCLIP}}(\text{VIC}, \hat{\text{VIC}}) \tag{2}$$

## 2.3 IMAGE CAPTIONING

To predict image captions from brain activity we convert the diffusion prior's predicted ViT-bigG/14 embeddings to CLIP ViT/L-14 space and feed those outputs through a frozen pretrained GenerativeImage2Text (GIT) model Wang et al. (2022). The use of GIT to caption images from brain

activity in the Natural Scenes Dataset was previously shown to be viable by Ferrante et al. (2023b). We independently trained a linear model to convert from OpenCLIP ViT-bigG/14 embeddings to CLIP ViT-L/14 embeddings (see Appendix A.9), which was necessary because there was no existing GIT model that accepted OpenCLIP ViT-bigG/14 embeddings as inputs. Image caption prediction from brain activity lends further flexibility to such decoding approaches and can help refine image reconstructions to match desired semantic content.

## 2.4 FINE-TUNING STABLE DIFFUSION XL FOR UNCLIP

CLIP Radford et al. (2021) is an example of a multimodal contrastive model that maps images and text captions to a shared embedding space. unCLIP (or image variations) models go from this shared embedding space back to pixel space, and have been used for the creative application of returning variations of a given reference image. As such, previous unCLIP models prioritized replication of high-level semantics over low-level structures. These models can be trained by fine-tuning a base image generation model to accept CLIP image embeddings instead of, or in addition to, text embeddings. Outputs are diffused from pure noise just like the base model, unlike image-to-image models Meng et al. (2022) that start the diffusion process from a reference image mixed with noise.

Contrary to previous unCLIP models, our goal was to train a model that returns images as close as possible to the reference image across both low-level structure and high-level semantics. This is because our use-case was to exactly return the original image given its CLIP image embedding predicted from the brain.

The base Stable Diffusion XL (SDXL) Podell et al. (2023) model uses text conditionings from both OpenCLIP ViT-bigG/14 and CLIP ViT-L/14. They condition cross-attention layers on the penultimate text encoder outputs and additionally condition on pooled text embeddings from OpenCLIP ViT-bigG/14 by adding it to the timestep embedding. Here, we fine-tuned the cross-attention layers using the OpenCLIP ViT-bigG/14 image embeddings corresponding to all 256 patch tokens and we dropped the additional conditioning on pooled text embeddings. We opted to only condition on image embeddings because we observed that incorporating any text conditioning worsened the fidelity of the unCLIP reconstructions.

We evaluate the fidelity of our SDXL unCLIP model to reconstruct images from ground truth OpenCLIP ViT-bigG/14 image embeddings in Appendix A.8, showing that reconstructions are nearly identical to the original images. We fine-tuned SDXL on one 8xA100 80GB GPU node using an internal dataset for $110,000$ optimization steps at a resolution of $256 \times 256$ pixels and a batch size of 8 with offset-noise Lin et al. (2024); Guttenberg (2023) set to $0.04$. All other settings were identical to those used with base Stable Diffusion XL. Like Stable Diffusion XL, this unCLIP model can output different aspect ratios, however, we observed best results with $768 \times 768$ resolution.

## 2.5 MODEL INFERENCE

The pipeline for reconstruction inference is depicted in Figure 2. First, the diffusion prior's predicted OpenCLIP ViT-bigG/14 image latents are fed through our SDXL unCLIP model to output a pixel image. We observed that these reconstructions were often distorted ("unrefined") due to an imperfect mapping to bigG space (see Figure **??**). This may be explained by the increased versatility allowed from mapping to the larger dimensionality OpenCLIP bigG latent space. To increase image realism, we feed the unrefined reconstructions from SDXL unCLIP through base SDXL via image-to-image Meng et al. (2022) with text conditioning guidance from MindEye2's predicted image captions (section 2.3). We skip the first 50% of denoising diffusion timesteps, starting the process from the noised image encoding of the unrefined reconstruction. We simply take the first samples output from these stochastic models without any special 2nd-order selection. Refinement using base SDXL subjectively improves the quality of image outputs without strongly affecting low or high-level image metrics.

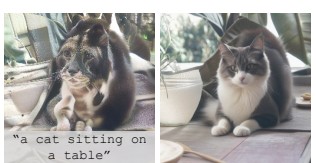

Figure 3: SDXL unCLIP reconstructions + predicted image captions (left) are fed to base SDXL for refinement (right).

The final "refined" reconstructions come from combining the outputs from base SDXL with the pixel images output from the low-level submodule via simple weighted averaging (4:1 ratio). This weighted averaging step increases performance on low-level image metrics while minimally affecting reconstructions' subjective appearance.

For retrieval inference, only the retrieval submodule's outputs are necessary. Nearest neighbor retrieval can be performed via cosine similarity between the submodule's OpenCLIP ViT-bigG/14 embeddings and all the ViT-bigG/14 embeddings corresponding to the images in the desired image pool.

## 3 RESULTS

We used the Natural Scenes Dataset (NSD) Allen et al. (2022), a public fMRI dataset containing the brain responses of human participants viewing rich naturalistic stimuli from COCO Lin et al. (2014). The dataset spans 8 subjects who were each scanned for 30-40 hours (30-40 separate scanning sessions), where each sesssion consisted of viewing 750 images for 3 seconds each. Images were seen 3 times each across the sessions and were unique to each subject, except for a select 1,000 images which were seen by all the subjects. We follow the standardized approach to train/test splits used by other NSD reconstruction papers Takagi and Nishimoto (2022); Ozcelik and VanRullen (2023); Gu et al. (2023) which is to use the shared images seen by all the subjects as the test set. We follow the standard of evaluating model performance across low- and high-level image metrics averaged across the 4 subjects who completed all 40 scanning sessions. We averaged across same-image repetitions for the test set (1,000 test samples) but not the training set (30,000 training samples). For more information on NSD and data preprocessing see Appendix A.1.

Critically, models trained on a subset of data were selected in chronological order. That is, models trained from only 1 hour's worth of data come from using the subject's first scanning session of 750 image presentations. This means our model must be able to generalize to test data collected from scanning sessions entirely held-out during training.

### 3.1 FMRI-TO-IMAGE RECONSTRUCTION

First, we report performance of MindEye2 when training on the full NSD dataset. We quantitatively compare reconstructions across fMRI-to-image models in Table 1, demonstrating state-of-the-art MindEye2 performance across nearly all metrics. We compare to both the previous MindEye1 results as well as other fMRI-to-image approaches that were open-sourced such that we could replicate their pipelines using the recently updated NSD (which includes an additional 3 scanning sessions for every subject).

MindEye2 refined reconstructions using the full NSD dataset performed SOTA across nearly all metrics, confirming that our changes to shared-subject modeling, model architecture, and training procedure benefitted reconstruction and retrieval performance (explored more in section 3.5). Interestingly, we observed that high-level metrics for the unrefined MindEye2 reconstructions outperformed the refined reconstructions across several metrics despite looking visibly distorted. This suggests that the standard evaluation metrics used across fMRI-to-image papers should be further scrutinized as they may not accurately reflect subjective interpretations of reconstruction quality.

We also report performance for MindEye2 fine-tuned with only 1 hour of data in the same Table 1. We qualitatively compare reconstructions side-by-side with models trained on only 1 hour's worth of data in Figure 4, depicting improvements in reconstruction quality for MindEye2. We report more evaluations in the Appendix: see A.3 for MindEye2 results without pretraining, A.6 for evaluations with varying amounts of training data across all models, A.7 for single-subject evaluations, and A.12 for MindEye2 evaluations with varying selection of pretraining subjects.

#### 3.1.1 VARYING AMOUNTS OF TRAINING DATA

The overarching goal of the present work is to showcase high-quality reconstructions of seen images from a single visit to an MRI facility. Figure 5 shows reconstruction performance across MindEye2 models trained on varying amounts of data from subject 1. There is a steady improvement across both pretrained and non-pretrained models as more data is used to train the model. "Non-pretrained" refers to single-subject models trained from scratch. The pretrained and non-pretrained results became

Seen image          Reconstructions using 1 hour of training data

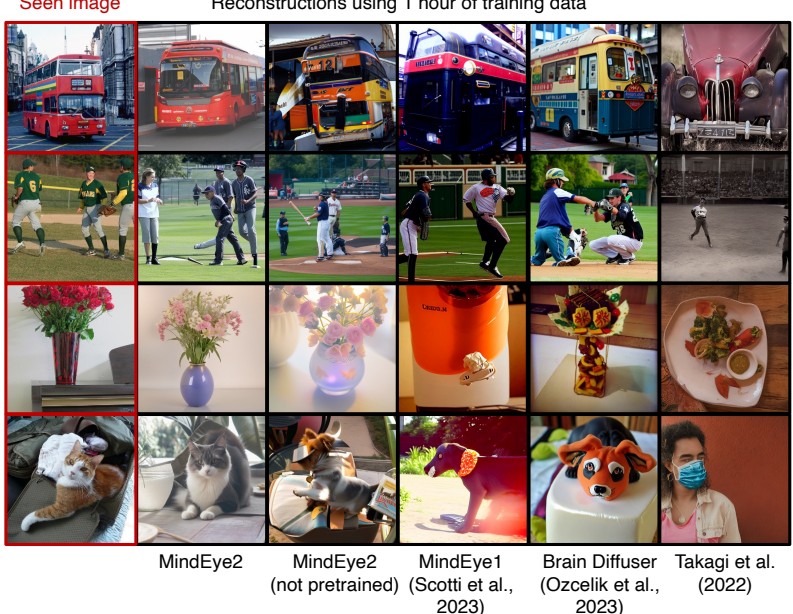

MindEye2    MindEye2         MindEye1            Brain Diffuser    Takagi et al.
            (not pretrained)  (Scotti et al.,    (Ozcelik et al.,  (2022)
                              2023)              2023)

Figure 4: Reconstructions from different model approaches using 1 hour of training data from NSD.

| Method | Low-Level | | | | High-Level | | | | Retrieval | |
|---|---|---|---|---|---|---|---|---|---|---|
| | PixCorr ↑ | SSIM ↑ | Alex(2) ↑ | Alex(5) ↑ | Incep ↑ | CLIP ↑ | Eff ↓ | SwAV ↓ | Image ↑ | Brain ↑ |
| MindEye2 | **0.322** | **0.431** | **96.1%** | 98.6% | 95.4% | 93.0% | **0.619** | 0.344 | **98.8%** | **98.3%** |
| MindEye2 (unrefined) | 0.278 | 0.328 | 95.2% | **99.0%** | **96.4%** | **94.5%** | 0.622 | **0.343** | – | – |
| MindEye1 | 0.319 | 0.360 | 92.8% | 96.9% | 94.6% | 93.3% | 0.648 | 0.377 | 90.0% | 84.1% |
| Ozcelik and VanRullen (2023) | 0.273 | 0.365 | 94.4% | 96.6% | 91.3% | 90.9% | 0.728 | 0.421 | 18.8% | 26.3% |
| Takagi and Nishimoto (2023) | 0.246 | 0.410 | 78.9% | 85.6% | 83.8% | 82.1% | 0.811 | 0.504 | – | – |
| MindEye2 (low-level) | 0.399 | 0.539 | 70.5% | 65.1% | 52.9% | 57.2% | 0.984 | 0.673 | – | – |
| MindEye2 (1 hour) | 0.195 | 0.419 | 84.2% | 90.6% | 81.2% | 79.2% | 0.810 | 0.468 | 79.0% | 57.4% |

Table 1: Quantitative comparison of fMRI-to-image models. Results from all previous work were recalculated using their respective public codebases using the full 40 sessions of NSD data, which was not released until the recent completion of the 2023 Algonauts challenge. Image retrieval refers to the percent of the time the correct image was retrieved out of 300 candidates, given the associated brain sample (chance=0.3%); vice-versa for brain retrieval. PixCorr=pixelwise correlation between ground truth and reconstructions; SSIM=structural similarity index metric Wang et al. (2004); EfficientNet-B1 ("Eff") Tan and Le (2020) and SwAV-ResNet50 ("SwAV") Caron et al. (2021) refer to average correlation distance; all other metrics refer to two-way identification (chance = 50%). Two-way identification refers to percent correct across comparisons gauging if the original image embedding is more similar to its paired brain embedding or a randomly selected brain embedding (see Appendix A.11). Missing values are from metrics being non-applicable. Bold indicates best performance, underline second-best performance.

increasingly more similar as more data was added. The 1-hour setting offers a good balance between scan duration and reconstruction performance, with notable improvements from pretraining. The non-pretrained models trained with 10 or 30 minutes of data suffered significant instability. These models may have experienced mode collapse where outputs were similarly nonsensical regardless of input. Such reconstructions coincidentally performed well on SSIM, indicating SSIM may not be a fully representative metric.

## 3.2 IMAGE CAPTIONING

Predicted image captions are quantitatively compared to previous work in Appendix A.2. UniBrain Mai and Zhang (2023) was first to predict captions using NSD, training a diffusion model to predict CLIP ViT-L/14 text latents which get fed through a pretrained Optimus GPT2 model Radford et al.

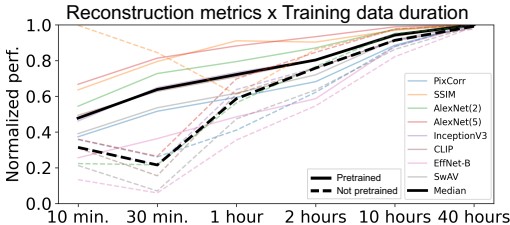

Figure 5: Normalized reconstruction metrics for MindEye2 with (connected) or without (dotted) pretraining on other subjects, using varying amounts of training/fine-tuning data. Normalization was such that 0 on the y-axis corresponds to metrics using random COCO images (not from NSD test set) as reconstructions and 1 corresponds to metrics using 40-session pretrained MindEye2. Black lines indicate median. Test data is the same across all comparisons (see section 3).

(2019). Ferrante et al. (2023b) predicted image captions by mapping fMRI inputs to CLIP ViT-L/14 image latents via ridge regression, passing these latents through a pretrained GIT model Wang et al. (2022). MindEye2 captioning performance outperformed previous models across all metrics except one, suggesting high-quality image captions from brain activity.

## 3.3 IMAGE/BRAIN RETRIEVAL

Image retrieval metrics help quantify the level of fine-grained image information contained in the fMRI embeddings. There are many images in the test set that contain similar semantic content (e.g., 14 images of zebras), so if the model can identify the exact image corresponding to a given brain sample, that demonstrates such fMRI embeddings contain fine-grained image content. MindEye2 improves upon MindEye1's retrieval evaluations by reaching near-ceiling performance on the retrieval benchmarks used in previous papers Lin et al. (2022); Scotti et al. (2023) (Table 1). Further, retrieval performance remained competitive when MindEye2 was trained with only 1 hour of data. Computing the retrieval metrics in Table 1 involved the steps described in Appendix A.10.

## 3.4 BRAIN CORRELATION

To measure whether a reconstruction is faithful to the original brain activity that evoked it, we examine whether it accurately predicts that brain activity when input to a encoding model pretrained to predict brain activity from images Gaziv et al. (2022). Encoding models provide a more comprehensive analysis of the proximity between images and brain activity Naselaris et al. (2011), providing a unique measure of reconstruction quality that is perhaps more informative than the image metrics traditionally used for assessment. This alignment is measured independently of the stimulus image, allowing it to be used to assess reconstruction quality when the ground-truth image is unknown, making it extendable to new data in a variety of domains including covert visual content such as mental images. Given that human judgment is grounded in human brain activity, it could also be the case that brain correlation metrics provide increased alignment with the judgments of human observers. The brain correlation metrics below are calculated with the GNet encoding model St-Yves et al. (2022) using protocol from Kneeland et al. (2023c). "Unrefined" reconstructions performed best, perhaps because refinement sacrifices brain alignment (and reconstruction performance as assessed by some metrics) for the additional boost in perceptual alignment from enforcing a naturalistic prior.

| Brain Region | MindEye2 | MindEye2 (unrefined) | MindEye2 (1 hour) | Brain Diffuser | Takagi et al. |
|---|---|---|---|---|---|
| Visual cortex↑ | 0.373 | **0.384** | 0.348 | 0.381 | 0.247 |
| V1↑ | 0.364 | **0.385** | 0.309 | 0.362 | 0.181 |
| V2↑ | 0.352 | **0.366** | 0.314 | 0.340 | 0.152 |
| V3↑ | 0.342 | **0.353** | 0.315 | 0.332 | 0.152 |
| V4↑ | 0.327 | **0.339** | 0.300 | 0.323 | 0.170 |
| Higher vis.↑ | 0.368 | 0.373 | 0.351 | **0.375** | 0.288 |

Table 2: Brain correlation scores calculated in different brain regions including visual cortex, early visual cortical regions V1, V2, V3, and V4, and higher visual areas (set complement of visual cortex and early visual cortex).

### 3.5 ABLATIONS

Here we explain where MindEye2 improvements over MindEye1 come from through ablations. MindEye2 outperforms MindEye1 even without pretraining on other subjects (see Appendix A.3), suggesting improvements in model architecture and training procedure. Our ablation results compare models trained from scratch in reduced capacity (1024-dim shared-subject latent space), skipping base SDXL refinement, using 10 sessions of data solely from subject 1.

Two core differences between MindEye2 and MindEye1 are (1) we used a linear layer, rather than an MLP with dropout, for the initial mapping of voxels to the dimensionality of the residual MLP backbone, and (2) we map to OpenCLIP bigG image latents rather than CLIP L latents. Our ablations in Appendix A.4 show that these changes improve performance across all metrics, suggesting that a linear layer with L2 regularization is a more effective means of initially mapping voxels into model space, and that bigG is the richer, more effective CLIP space to map fMRI activity into.

Ablations in Appendix A.5 show evaluations from models trained with various combinations of components. Retrieval metrics were worst when MindEye2 was trained with the diffusion prior and low-level submodules removed, and reconstruction metrics were worst when trained with the retrieval submodule and low-level submodule removed. More fine-grained comparison reveals that reconstruction metrics benefitted substantially from the retrieval submodule, although the low-level submodule did not considerably impact performance.

## 4 RELATED WORK

It is common for fMRI analyses to align subjects' brains to a shared space for the purposes of increasing statistical power and/or assessing generality of scientific findings. Such alignment is difficult because structural and functional topography differs substantially across people Talairach and Tournoux (1990); Mazziotta et al. (2001). There are many approaches to functional alignment but typically they involve subjects experiencing shared stimuli and then using responses to these stimuli to learn an alignment mapping Chen et al. (2015); Haxby et al. (2011); Huang et al. (2021); Nastase et al. (2019); Busch et al. (2021). While it is useful to conduct such experiments to identify sources of shared signal across subjects, it is also limiting in that new subjects would need to be scanned using the same experimental protocol. Other functional alignment approaches avoid such limitations by using self-supervised learning to identify an initial generalizable embedding space with outputs suitable for downstream tasks Schneider et al. (2023); Chen et al. (2023a;b). Closest to our alignment approach are models that adopt both shared-subject and subject-specific mappings in their model architecture Défossez et al. (2022); Benchetrit et al. (2023); Yang et al. (2023).

Ferrante et al. (2023a) previously showed across-subject image reconstruction via ridge regression by training a linear subject-specific decoding model and then separately mapping other subjects to this space via ridge regression. This is similar to our approach in that both involve ridge regression to a shared space, but is distinct in that their approach is capped by the performance of the initial single-subject model from which other subjects are mapped into, is restricted to only linear fine-tuning, and was demonstrated only with a reduced training dataset of images seen by all subjects. MindEye2 is unique in its demonstration that a single neural network model can be pretrained across subjects experiencing unique stimuli and robustly fine-tuned to a new subject with few data points.

## 5 CONCLUSION

We introduce MindEye2, a modeling approach that outputs reconstructions of seen images from fMRI activity with a similar quality to previous approaches using only a fraction of the training data. MindEye2 further achieves SOTA across reconstruction and retrieval metrics when supplied with the full training data. Our approach pretrains a model using data from multiple subjects, which is then fine-tuned on scarce data from a held-out subject. Patterns of fMRI activity are mapped to CLIP space and images are reconstructed with the help of our unCLIP model fine-tuned from Stable Diffusion XL. Our work shows the potential to apply deep learning models trained on large-scale neuroimaging datasets to new subjects with minimal data.

The present work demonstrates that it is now practical for patients to undergo a single MRI scanning session and produce enough data to perform high-quality reconstructions of their visual perception.

Such image reconstructions from brain activity are expected to be systematically distorted due to factors including mental state, neurological conditions, etc. This could potentially enable novel clinical diagnosis and assessment approaches, including applications for improved locked-in (pseudocoma) patient communication Monti et al. (2010) and brain-computer interfaces if adapted to real-time analysis Wallace et al. (2022) or non-fMRI neuroimaging modalities. Some limitations of this work are that fMRI is extremely sensitive to movement and requires subjects to comply with the task: decoding is easily resisted by slightly moving one's head or thinking about unrelated information Tang et al. (2023). MindEye2 has also only been shown to work on natural scenes such as those in COCO; additional data and/or specialized generative models would likely be required for other image distributions. As technology continues to improve, we note it is important that brain data be carefully protected and companies collecting such data be transparent with their use.

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

# A    APPENDIX

## A.1    ADDITIONAL DATASET INFORMATION

fMRI responses correspond to normalized single-trial betas output from GLMSingle Prince et al. (2022). We use preprocessed flattened fMRI voxels in 1.8-mm native volume space corresponding to the "nsdgeneral" brain region, defined by the NSD authors as the subset of voxels in posterior cortex most responsive to the visual stimuli presented (between 13,000 to 16,000 voxels per participant). MindEye2 was developed using a training and test set of subject 1's data, with other subjects' data untouched until final training of models. The fMRI data from both the training and test set was normalized using a voxel-wise Z-scoring procedure using the mean and standard deviation calculated using only the training set. Despite the shared1000 test trials being distributed across the scanning sessions for each subject, we chose to keep the test set consistent no matter the number of sessions being used for training. We also adjusted the number of training sessions after the normalization step, allowing us to keep the statistical properties of the shared1000 test set consistent between experiments with varying amounts of training data. This may inadvertently give a small normalization advantage to models trained with fewer training sessions, as the models are normalized with additional data not made available for training.

## A.2    IMAGE CAPTIONING

We adopt the same caption metrics reported in the previous work. ROUGE Lin (2004) and METEOR Banerjee and Lavie (2005) capture aspects of text structure and composition. CLIP Radford et al. (2021) and SentenceTransformer ("all-MiniLM-L6-v2") Reimers and Gurevych (2020) are higher-level metrics that provide insight into textual context, relationships, and semantics. All metrics except ROUGE were calculated using the same code as Ferrante et al. (2023b).

| | COCO captions | | GIT captions | |
|---|---|---|---|---|
| Metric | MindEye2 | UniBrain | MindEye2 | Ferrante et al. |
| METEOR ↑ | **0.248** | 0.170 | **0.344** | 0.305 |
| ROUGE-L ↑ | **0.326** | 0.225 | **0.427** | - |
| ROUGE-1 ↑ | **0.353** | 0.247 | **0.455** | - |
| Sentence ↑ | **47.9%** | - | **52.3%** | 44.7% |
| CLIP-B ↑ | **73.7%** | - | **75.4%** | 70.5% |
| CLIP-L ↑ | 63.8% | **86.1%** | **67.1%** | - |

Table 3: FMRI-to-image caption evaluations. Previous works used different ground truth captions for comparison (COCO captions or captions generated from GIT), necessitating separate comparisons. Results were calculated exclusively on NSD subject 1. MindEye2 metrics come from the model trained on all 40 sessions of NSD data whereas previous work used 37 sessions.

## A.3    MINDEYE2 (NOT PRETRAINED) VS. MINDEYE1

Table 4 shows how MindEye2 outperforms MindEye1 even without pretraining on other subjects. Models were trained using the full 40 sessions of training data from subject 1. This suggests that improvements from MindEye1 to MindEye2 are not explained solely from pretraining on other subjects, but that benefits also come from improved model architecture and training procedure.

## A.4    ABLATION: MINDEYE2 CORE CHANGES FROM MINDEYE1

Ablations results in Table 5 compare MindEye2 results if training the model using MindEye1's MLP with dropout or MindEye1's mapping to CLIP ViT-L/14 space instead of OpenCLIP ViT-bigG/14 space.

## A.5    ABLATION: MINDEYE2 MODEL COMPONENTS

Ablations results in Table 6 compare MindEye2 results if training the model with certain components removed.

| Method | | MindEye2 | MindEye1 |
|---|---|---|---|
| Low-Level | PixCorr ↑ | 0.376 | **0.388** |
| | SSIM ↑ | **0.440** | 0.355 |
| | Alex(2) ↑ | **97.5%** | 96.1% |
| | Alex(5) ↑ | **99.1%** | 98.3% |
| High-Level | Incep ↑ | **95.4%** | 95.0% |
| | CLIP ↑ | 92.6% | **93.7%** |
| | Eff ↓ | **0.612** | 0.635 |
| | SwAV ↓ | **0.341** | 0.360 |
| Retrieval | Fwd ↑ | **100.0%** | 95.0% |
| | Bwd ↑ | **99.7%** | 89.4% |
| Brain Corr | NSD General ↑ | **0.370** | 0.353 |
| | V1 ↑ | **0.383** | 0.349 |
| | V2 ↑ | **0.373** | 0.336 |
| | V3 ↑ | **0.363** | 0.328 |
| | V4 ↑ | **0.335** | 0.307 |
| | Higher vis. ↑ | **0.356** | 0.345 |

Table 4: Performance comparison between MindEye2 (refined) and MindEye1 both trained from scratch across all 40 NSD sessions using only subject 1 data.

| Metric | | ME2 | ME1 | CLIP L |
|---|---|---|---|---|
| Low-Level | PixCorr ↑ | **0.292** | 0.225 | 0.243 |
| | SSIM ↑ | **0.386** | 0.380 | 0.371 |
| | Alex(2) ↑ | **92.7%** | 87.3% | 84.8% |
| | Alex(5) ↑ | **97.6%** | 94.7% | 93.7% |
| High-Level | Incep ↑ | **91.5%** | 88.9% | 87.7% |
| | CLIP ↑ | **90.5%** | 86.2% | 89.2% |
| | Eff ↓ | **0.700** | 0.758 | 0.744 |
| | SwAV ↓ | **0.393** | 0.430 | 0.427 |
| Retrieval | Fwd ↑ | **97.4%** | 84.9% | 89.6% |
| | Bwd ↑ | **95.1%** | 70.6% | 82.8% |

Table 5: Ablations on how MindEye2 (ME2) improves upon MindEye1. "ME1" results replace the initial linear mapping of fMRI voxels with MindEye1's MLP with dropout. "CLIP L" results map voxels to CLIP L (reconstructions via Versatile Diffusion) instead of OpenCLIP bigG (reconstructions via SDXL unCLIP).

| Metric | | Prior | Prior+Low | Prior+Ret. | All |
|---|---|---|---|---|---|
| Low-Level | PixCorr ↑ | 0.155 | **0.281** | 0.233 | 0.267 |
| | SSIM ↑ | 0.309 | 0.385 | 0.319 | **0.380** |
| | Alex(2) ↑ | 79.6% | 89.4% | **90.6%** | 89.7% |
| | Alex(5) ↑ | 88.6% | 96.2% | **96.8%** | 96.4% |
| High-Level | Incep ↑ | 85.3% | 91.5% | **91.9%** | 91.4% |
| | CLIP ↑ | 79.5% | 88.4% | **89.4%** | 87.9% |
| | Eff ↓ | 0.805 | 0.727 | **0.717** | 0.732 |
| | SwAV ↓ | 0.490 | 0.416 | **0.410** | 0.415 |
| | | Ret. | Ret.+Low | Prior.+Ret. | All |
| Retrieval | Fwd ↑ | 96.5% | 96.9% | 96.2% | **98.0%** |
| | Bwd ↑ | 92.4% | 93.0% | **95.8%** | 94.1% |

Table 6: Ablations compare reconstruction and retrieval metrics for MindEye2 trained with various combinations of model components. Retr.=Retrieval submodule, Low=Low-level submodule.

## A.6 RECONSTRUCTION EVALUATIONS ACROSS VARYING AMOUNTS OF TRAINING DATA

Here we present a further analysis of how model performance scales with training data. All of the results presented in Figures 6, 7, and 8 are calculated on only subject 1.

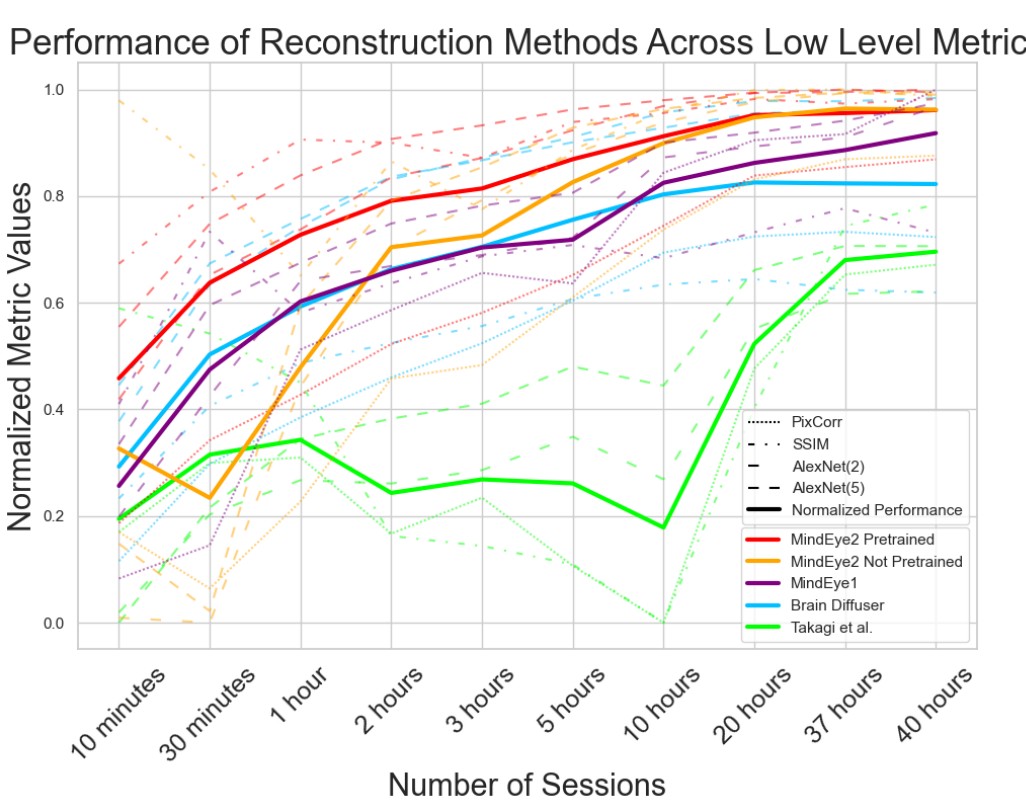

Figure 6: Low-level metric performance (y-axis) plotted against the number of fMRI scanning sessions used in the training data (x-axis) for subject 1. All values are normalized to the same y-axis. The bolded line represents the average performance across all metrics.

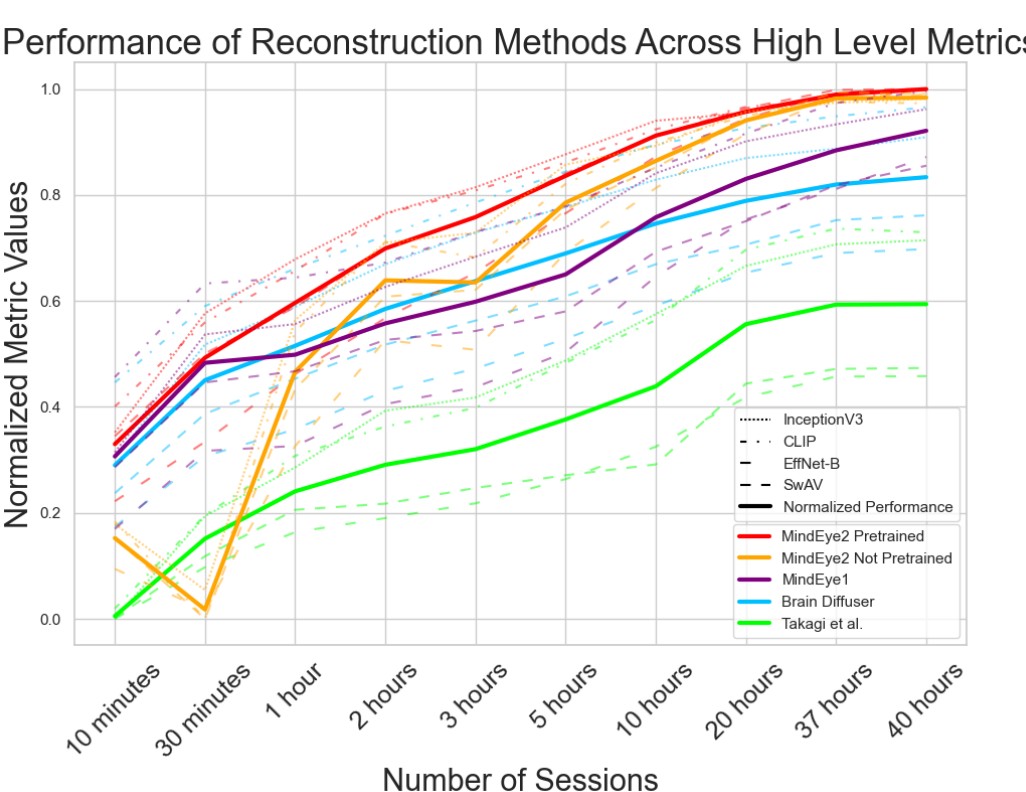

Figure 7: High-level metric performance (y-axis) plotted against the number of fMRI scanning sessions used in the training data (x-axis) for subject 1. All values are normalized to the same y-axis. The bolded line represents the average performance across all metrics. SwAV and EffNet-B scores are inverted in this plot so that higher is better for all metrics.

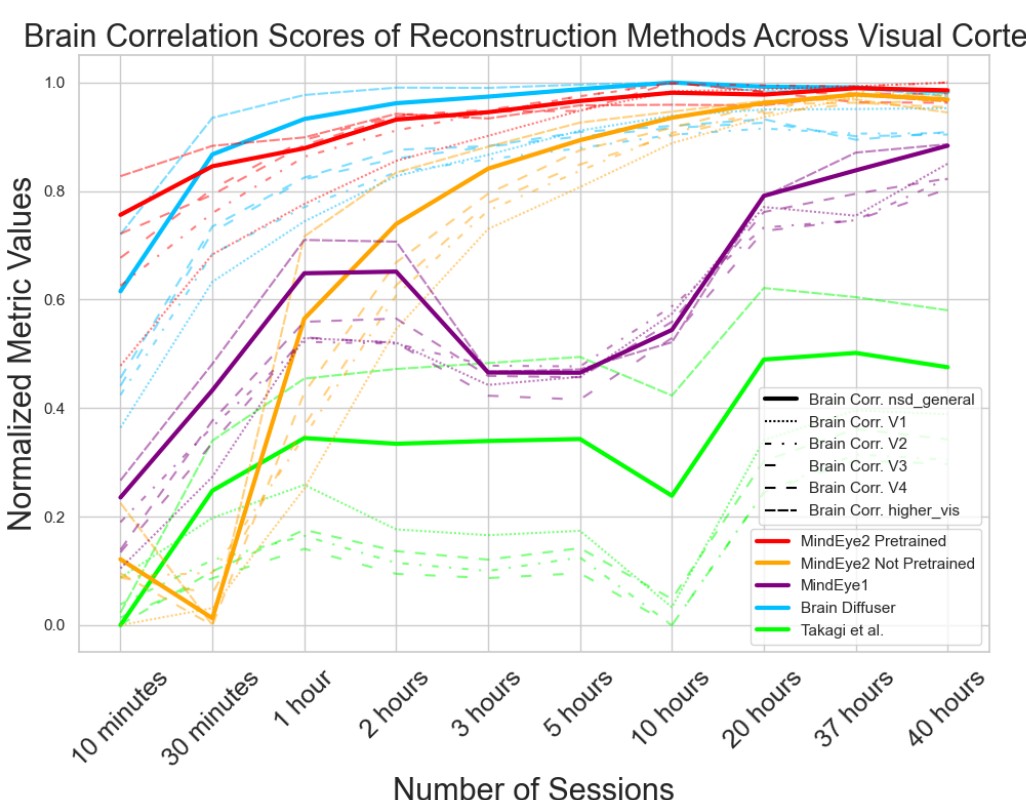

Figure 8: Brain correlation scores (y-axis) in different brain regions including visual cortex (defined by the nsdgeneral mask, bolded), V1, V2, V3, V4 (collectively called early visual cortex) and higher visual areas (the set complement of nsdgeneral and early visual cortex) plotted against the number of fMRI scanning sessions used in the training data (x-axis) for subject 1. All values are normalized to the same y-axis.

## A.7 SINGLE-SUBJECT EVALUATIONS

Tables 7 and 8 show more exhaustive evaluation metrics computed for every subject individually using 40-hours and 1-hour of fine-tuning data respectively.

| 40 Session Subject Results | | Subject 1 | Subject 2 | Subject 5 | Subject 7 |
|---|---|---|---|---|---|
| Low | PixCorr ↑ | **0.374** | 0.328 | 0.301 | 0.283 |
| | SSIM ↑ | **0.439** | 0.430 | 0.432 | 0.423 |
| | Alex(2) ↑ | **97.82%** | 97.01% | 95.32% | 94.25% |
| | Alex(5) ↑ | **99.10%** | 98.83% | 98.54% | 97.97% |
| High | Incep ↑ | 96.15% | 94.90% | **96.52%** | 94.09% |
| | CLIP ↑ | 93.56% | 91.66% | **94.32%** | 92.36% |
| | Eff ↓ | 0.609 | 0.631 | **0.600** | 0.638 |
| | SwAV ↓ | 0.338 | 0.347 | **0.335** | 0.357 |
| Retrieval | Image ↑ | **99.96%** | 99.88% | 98.39% | 96.89% |
| | Brain ↑ | **99.87%** | 99.84% | 96.94% | 96.53% |
| Brain Corr. | Visual cortex ↑ | 0.374 | 0.387 | **0.413** | 0.317 |
| | V1 ↑ | 0.389 | **0.391** | 0.354 | 0.321 |
| | V2 ↑ | **0.381** | 0.353 | 0.359 | 0.314 |
| | V3 ↑ | **0.367** | 0.362 | 0.340 | 0.299 |
| | V4 ↑ | 0.337 | **0.374** | 0.321 | 0.278 |
| | Higher vis. ↑ | 0.361 | 0.380 | **0.424** | 0.309 |
| Captions | METEOR ↑ | 0.248 | 0.245 | **0.250** | 0.240 |
| | ROUGE-L ↑ | 0.326 | 0.321 | **0.327** | 0.319 |
| | ROUGE-1 ↑ | 0.353 | 0.349 | **0.354** | 0.347 |
| | Sentence ↑ | 47.95% | 46.69% | **49.40%** | 46.97% |
| | CLIP-B ↑ | 73.74% | 73.15% | **74.22%** | 73.16% |
| | CLIP-L ↑ | 63.76% | 62.96% | **64.14%** | 62.86% |

Table 7: Single subject quantitative results for 40 sessions of training data.

| 1 Session Subject Results | | Subject 1 | Subject 2 | Subject 5 | Subject 7 |
|---|---|---|---|---|---|
| Low | PixCorr ↑ | **0.235** | 0.200 | 0.175 | 0.170 |
| | SSIM ↑ | 0.428 | **0.433** | 0.405 | 0.408 |
| | Alex(2) ↑ | 88.02% | **85.00%** | 83.11% | 80.70% |
| | Alex(5) ↑ | **93.33%** | 92.13% | 91.00% | 85.90% |
| High | Incep ↑ | 83.56% | 81.86% | **84.33%** | 74.90% |
| | CLIP ↑ | 80.75% | 79.39% | **82.53%** | 74.29% |
| | Eff ↓ | 0.798 | 0.807 | **0.781** | 0.854 |
| | SwAV ↓ | 0.459 | 0.467 | **0.444** | 0.504 |
| Retrieval | Image ↑ | **93.96%** | 90.53% | 66.94% | 64.44% |
| | Brain ↑ | **77.63%** | 67.18% | 46.96% | 37.77% |
| Brain Correlation | Visual cortex ↑ | 0.347 | 0.350 | **0.404** | 0.294 |
| | V1 ↑ | 0.318 | 0.306 | **0.328** | 0.283 |
| | V2 ↑ | **0.337** | 0.296 | 0.336 | 0.285 |
| | V3 ↑ | **0.341** | 0.323 | 0.323 | 0.272 |
| | V4 ↑ | 0.316 | **0.336** | 0.304 | 0.243 |
| | Higher vis. ↑ | 0.345 | 0.357 | **0.415** | 0.285 |
| Captions | METEOR ↑ | 0.200 | 0.200 | **0.207** | 0.189 |
| | ROUGE-L ↑ | 0.278 | 0.272 | **0.280** | 0.260 |
| | ROUGE-1 ↑ | 0.299 | 0.293 | **0.300** | 0.279 |
| | Sentence ↑ | 33.52% | 32.36% | **35.12%** | 28.00% |
| | CLIP-B ↑ | 67.22% | 65.98% | **67.63%** | 63.15% |
| | CLIP-L ↑ | 55.44% | 54.00% | **56.19%** | 50.60% |

Table 8: Single subject quantitative results for 1 session of training data.

## A.8 UNCLIP EVALUATION

Previous fMRI-to-image papers Scotti et al. (2023); Ozcelik and VanRullen (2023); Mai and Zhang (2023) opted for Versatile Diffusion because it was state-of-the-art in reconstructing images from CLIP image latents with little variation. To compare the image generation capabilities of our unCLIP model with Versatile Diffusion, we computed Fréchet inception distance (FID) Heusel et al. (2018) scores across 30,000 randomly sampled images from the COCO 2017 validation set. The images were center-cropped and scaled to $480 \times 480$ resolution. For Versatile Diffusion, we used Huggingface's VersatileDiffusionDualGuidedPipeline with text_to_image set to 0 to not take any input from text.

Our unCLIP model fine-tuned from Stable Diffusion XL outperforms Versatile Diffusion in terms of returning the original image from CLIP latents (see Appendix 9). This difference is visually obvious as shown in Figure 9. Note that while we observed distortions in our unrefined fMRI-to-image reconstructions using our unCLIP model fine-tuned from SDXL, such distortions were rare when using the ground truth CLIP embeddings.

The ability for this unCLIP model to nearly perfectly return the original image also indicates that OpenCLIP ViT-bigG image embeddings effectively preserve the majority of the information inherent in the original pixel image, retaining both low-level structure and high-level semantic details.

## A.9 OPENCLIP BIGG TO CLIP L CONVERSION

To map from OpenCLIP ViT-bigG/14 image latents to CLIP ViT-L/14 image latents during MindEye2 inference we independently trained a linear model using ground truth images from the COCO 2017 train and validation dataset. This conversion was necessary to use the pretrained GIT image captioning model. The PyTorch code used to train this model is depicted in Algorithm 1.

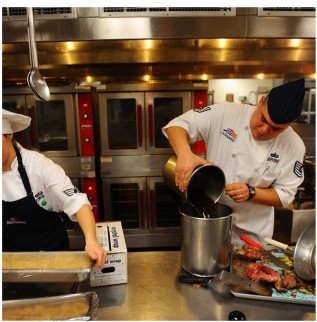
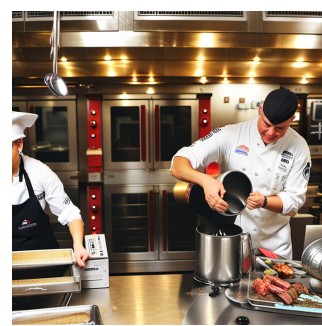
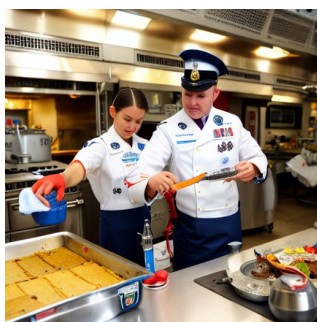

| Original image | SDXL unCLIP (OpenCLIP ViT-bigG/14) | Versatile Diffusion (CLIP ViT-L/14) |

Figure 9: Generating images from their CLIP image embeddings. SDXL unCLIP (middle) outperforms Versatile Diffusion (right) in capturing perceptual details.

| Metrics | SDXL unCLIP | VD |
|---|---|---|
| FID ↓ | **13.69** | 22.04 |
| PixCorr ↑ | **0.676** | 0.266 |
| SSIM ↑ | **0.232** | 0.055 |
| Alex(2) ↑ | **0.998** | 0.972 |
| Alex(5) ↑ | **0.998** | 0.966 |
| Incep ↑ | **0.997** | 0.994 |
| CLIP ↑ | **0.999** | 0.997 |
| Eff ↓ | **0.240** | 0.487 |
| SwAV ↓ | **0.029** | 0.108 |

Table 9: SDXL unCLIP reconstructions from ground truth OpenCLIP image latents consistently outperform Versatile Diffusion reconstructions from ground truth CLIP image latents.

## A.10   COCO RETRIEVAL

The goal for brain retrieval is to identify the correct sample of brain activity that gave rise to the seen image out of a pool of brain samples. The seen image is converted to an OpenCLIP image embedding (or CLIP image embedding, depending on the contrastive space used in the paper) and cosine similarity is computed between its respective fMRI latent (e.g., from the retrieval submodule) as well as 299 other randomly selected fMRI latents in the test set. For each test sample, success is determined if the cosine similarity is greatest between the ground truth OpenCLIP/CLIP image embedding and its respective fMRI embedding (aka top-1 retrieval performance, chance=1/300). We specifically used 300 random samples because this was the approach used in previous work. We averaged retrieval performance across test samples and repeated the entire process 30 times to account for the variability in random sampling of batches. For image retrieval, the same procedure is used

**Algorithm 1** PyTorch code to convert OpenCLIP bigG to CLIP L.

```
class BigG_to_L(torch.nn.Module):
    def __init__(self):
        super(BigG_to_L, self).__init__()
        self.linear1 = nn.Linear(clip_seq_dim,
                                 clip_text_seq_dim)
        self.linear2 = nn.Linear(clip_emb_dim,
                                 clip_text_emb_dim)
    def forward(self, x):
        x = self.linear1(x)
        x = self.linear2(x.permute(0,2,1))
        return x
```

except image and brain samples are flipped such that the goal is to find the corresponding seen image in the image pool from the provided brain sample.

MindEye1 scaled up image retrieval using a pool of billions of image candidates contained in the LAION-5B dataset Schuhmann et al. (2022). This was possible because all LAION images were already converted to CLIP L embeddings and made available for nearest neighbor lookup via the CLIP Retrieval client Beaumont (2022). We were not able to use this approach for MindEye2 because it would require converting all images to the $256 \times 1664$ dimensionality bigG latent space which was not feasible. That said, cursory investigation with the comparatively smaller MS-COCO dataset suggests that retrieval from a pool of images not containing the original image may not work as well with OpenCLIP bigG embeddings compared to the CLIP L embeddings used in MindEye1. To test retrieval, we used FAISS Douze et al. (2024) for k-nearest neighbor search through an index of flattened OpenCLIP bigG embeddings of 73,000 MS-COCO images. We found that for incorrect retrievals, the 3 nearest neighbors usually were dissimilar to the original image both semantically and in low-level appearance. This could be due to the latents corresponding to the 256 image patch tokens of OpenCLIP bigG representing a more complex combination of different levels of information. This could cause the OpenCLIP bigG embeddings to not be as effective for nearest neighbor retrieval in terms of subjective intepretation, as the last layer of CLIP ViT-L/14 is highly semantic but lacks in low-level image content. Although we demonstrated improved retrieval performance for MindEye2 compared to MindEye1 using random subsets of 300 images for MindEye2 compared to MindEye1 (Table 5), we suggest that mapping to the last layer of CLIP ViT-L/14 image space would work better if the intended application is to find semantically related nearest neighbors in a large image pool.

### A.11 RECONSTRUCTION EVALUATIONS: ADDITIONAL INFORMATION

Two-way comparisons were performed for AlexNet Krizhevsky et al. (2012) (second and fifth layers), InceptionV3 Szegedy et al. (2016) (last pooling layer), and CLIP (final layer of ViT-L/14). We followed the same image preprocessing and the same two-way identification steps as Ozcelik and VanRullen (2023) and Scotti et al. (2023). For two-way identification, for each model, we computed the Pearson correlation between embeddings for the ground truth image and the reconstructed image, as well as the correlation between the ground truth image and a different reconstruction elsewhere in the test set. If the correlation for the former was higher than the latter, this was marked as correct. For each test sample, performance was averaged across all possible pairwise comparisons using the other 999 reconstructions to ensure no bias from random sample selection. This yielded 1,000 averaged percent correct outputs, which we averaged across to obtain the metrics reported in Table 1.

### A.12 PRETRAINING WITH LESS SUBJECTS

To determine the relative impact of using additional subjects for pretraining, we separately fine-tuned a MindEye2 model for subject 1 (using 1 hour of their training data) that was pretrained only on subjects 2, 5, and 7 (these are the subjects who completed all 40 scanning sessions), as well as only on subject 5 (the subject whose single-subject model performed the best). Results in Table 10 show similar performance for these models compared to pretraining on the full set of available subjects, suggesting that the number of pretraining subjects does not seem to play a major role in subsequent fine-tuning performance.

### A.13 UMAP DIMENSIONALITY REDUCTION

As discussed in Scotti et al. (2023), UMAP dimensionality reduction McInnes et al. (2020) plots of disjointed CLIP fMRI embeddings next to aligned CLIP fMRI embeddings visualize how the diffusion prior effectively addresses the disjointed embedding spaces problem. Theoretically, multimodal contrastive learning will always produce disjointed embeddings because of the "modality gap" phenomenon whereby encoding modalities into a shared space restricts the effective embedding space to a narrow cone in geometric space Liang et al. (2022).

| | Metric | Sub 2-8 | Sub 2,5,7 | Sub 5 |
|---|---|---|---|---|
| Low-Level | PixCorr ↑ | **0.235** | 0.234 | 0.232 |
| | SSIM ↑ | **0.428** | 0.421 | 0.421 |
| | Alex(2) ↑ | 88.0% | **89.1%** | 88.9% |
| | Alex(5) ↑ | 93.3% | **94.1%** | 93.6% |
| High-Level | Incep ↑ | 83.6% | **83.8%** | **83.8%** |
| | CLIP ↑ | 80.8% | **83.5%** | 82.7% |
| | Eff ↓ | 0.798 | 0.790 | **0.787** |
| | SwAV ↓ | 0.459 | 0.448 | **0.447** |
| Retrieval | Fwd ↑ | **94.0%** | 92.7% | 90.6% |
| | Bwd ↑ | 77.6% | **80.8%** | 80.3% |
| Brain Corr | Visual cortex ↑ | 0.347 | **0.353** | 0.352 |
| | V1 ↑ | **0.318** | 0.316 | **0.318** |
| | V2 ↑ | **0.337** | 0.331 | 0.336 |
| | V3 ↑ | 0.341 | 0.339 | **0.342** |
| | V4 ↑ | 0.316 | **0.319** | 0.318 |
| | Higher vis. ↑ | 0.345 | **0.355** | 0.354 |

Table 10: Evaluation metrics for MindEye2 models both fine-tuned on 1 hour of training data from subject 1 but pretrained on different numbers of subjects.

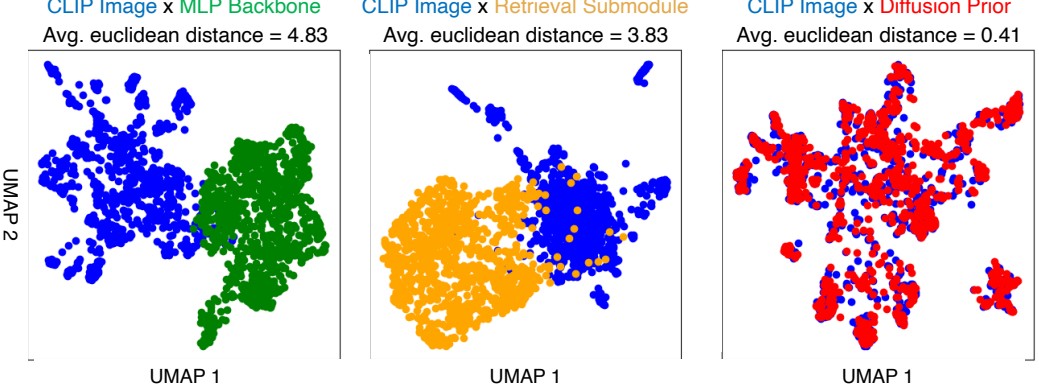

Figure 10: UMAP plots depict CLIP image latents (blue), backbone latents (green), retrieval sub-module latents (orange), and diffusion prior latents (red). UMAPs were estimated across the 1,000 test samples for subject 1, using the full 40-session model. CLIP image latents correspond to the $256 \times 1664$ dimensionality of OpenCLIP ViT-bigG/14 image token embeddings. Euclidean distance between the given MindEye2 embedding space and CLIP image space is lowest for the diffusion prior, suggesting that the diffusion prior helps to align the two embedding spaces.

