# OpenReview forum: "MindEye2: Shared-Subject Models Enable fMRI-To-Image With 1 Hour of Data"
_ICLR.cc/2024/Workshop/Re-Align — ICLR 2024 Workshop Re-Align Poster_

### Official Review · Reviewer_E6Gk · 2024-02-23
**Not the best aligned paper, but a pretty nice contribution nonetheless!**

**Rating:** 2
**Fit:** 2
**Confidence:** 1

**Workshop Review:**

Clarity 5/5
Correctness 5/5
Novelty 3/5
Interest/Relevance 2/5

The authors introduce MindEye2, a model that can reconstruct images from fMRI activity using only limited training data per subject. The model achieves state-of-the-art results on the Natural Scenes Dataset using full training data, and retains competitiveness with just 1 hour of data.

**Reason For Not Giving Higher Score:**

I am not sure that the paper is THAT well aligned with the theme of the workshop -- comparing biological and engineered representations -- and would have fit better into a brain-decoding/brain-ML workshop.

**Reason For Not Giving Lower Score:**

Yet, it is not too far removed from the ideas that form the central theme. The contribution is a new level of performance, the methods are thoroughly analyzed and the paper is very well written, and so I recommend acceptance.

**Reviewer Domain:**

machine learning

---

### Official Review · Reviewer_QzZy · 2024-02-24
**Awesome work!**

**Rating:** 3
**Fit:** 3
**Confidence:** 2

**Workshop Review:**

This work proposes a new version (MindEye2) of MindEye1, a model that can reconstruct a person’s visual perception during an fMRI scan. The new model both outperforms MindEye1 more generally, and
is able to generalize to a new subject after pre-training on other subjects, with only 1 hour of data from the new subject, as opposed to 40.

**Clarity** \
I think the paper is generally clearly written, and fairly easy to understand. The paper stands out on all other fronts, but this is the main point of improvement. Especially section 2, where the authors go over all of the submodules that make up their model. I believe that for a more general audience, it may be important to intuitively explain all of the steps in the model. The text comes across as rather jargon-y at times, and this can reduce the method’s interpretability for less familiar readers. Specifically, you use two CLIP models full names and it is hard for a reader to distinguish between the two (OpenCLIP ViT-bigG/14 and CLIP ViT-L/14) after reading about them many times. Moreover, much of this section reads as rather descriptive and detailed, whereas a more general reviewer is likely helped by a more intuitive explanation, rather than an exact explanation. You can refer to a more exact explanation in the Appendix if you don’t have enough space. A few examples are: P4 “We used the same BiMixCo and SoftCLIP losses used in MindEye1, which involved the first third of training iterations using bidirectional MixCo data augmentation and the last two-thirds of training iterations using soft labels…”. This sentence essentially assumes you have read Mindeye1, which is fine in the appendix, but I think the explanation can be more intuitive. Another example: P4 “The MLP projector feeds to a CNN upsampler that upsamples to be (64, 64, 3) dimensionality of SD VAE latents with L1 loss as well as an additional MLP to the embeddings of a teacher linear segmentation model VICRegL ConvNext-XXL for an auxilary SoftCLIP loss…”. This happens more often throughout the text, and apart from the run-on sentence structure sometimes, I believe the authors should focus more on why they did certain things and what the mechanistic advantage is.

**Correctness** \
The analyses in the paper are correct as far as I can tell, and so are the conclusions from their experiments. It is also commendable that the authors have made the code available online already.

**Novelty** \
I think the paper is novel in that it clearly improves on a previous model with ablations to show that each of their extensions have a positive impact, and the fact that they were able to figure out pre-training in this type of setting such that their model only requires 1 hour of fMRI data, which is quite unique.

**Interest to the community** \
I believe this paper is of very direct interest to the community.

**Grammatical/spelling errors (these may be personal preference so feel free to ignore them)** \
- P4, “auxillary” -> auxilliary
- P5 “We simply take the first samples output from these stochastic models…” -> first few samples from these
- P6, “We also report performance…” -> We also report the performance
- P6, “… of data suffered significant instability” -> suffered instabilities/significant instabilities
- P8, “… whether it accurately predicts that brain activity when input to a encoding model …” -> when used as input to an encoding model
- P8, “… using protocol from …” -> using the protocol from
- P8, “Here we explain where MindEye2 improvements over MindEye1 come from through ablations.” -> In this section we explain, with ablations, which additions lead to MindEye2’s improvement over MindEye1
- P11, the link in the reference at the bottom of the page goes out of screen on the right

**Reason For Not Giving Higher Score:**

N/A

**Reason For Not Giving Lower Score:**

I think this is an important piece of work to highlight at the workshop, and although it is not directly looking at the alignment of biological and artificial representations, it does showcase a method with which we can interpret and extract representations from noisy, indirect, and whole-brain measurements of the human brain. We can potentially compare the representational alignment to artificial neural networks in the future.

**Reviewer Domain:**

machine learning

---

### Official Review · Reviewer_tgi9 · 2024-02-26
**Improved sample efficiency in visual perception reconstruction from fMRI through inter-subject alignment**

**Rating:** 3
**Fit:** 3
**Confidence:** 1

**Workshop Review:**

This work aims to improve the sample efficiency in the reconstruction of visual perception from fMRI recordings. In particular, it proposes a pre-training heuristic for alignment of asynchronous fMRI responses from multiple individuals into a shared representational space, coupled with a mapping to CLIP space. Images are reconstructed with the help of an "unCLIP" model. After fine-tuning to a new subject with limited data, the procedure achieves good generalisation to held-out data from that subject across several metrics.

The proposed method is simple and the findings appear interesting. My main reservation is that some aspects of this work may be explained more clearly. The method introduced here has several advantages with respect to standard subject-alignment methods for neuroimaging, where "new subjects would need to be scanned using the same experimental protocol"; it is however worth remarking that those usually come with statistical guarantees, and that the proposed method is rather heuristic in comparison. Sections 2.2.1 to 2.2.3 are not easy to parse for readers without the required prior knowledge.

> The retrieval submodule is contrastively trained and produces disjointed CLIP-fMRI embeddings that can be used to find the original (or nearest neighbor) image in a pool of images, but is not used to reconstruct a novel image

It would be interesting to have more discussion of pros and cons, and the respective importance, of retrieval vs. reconstruction tasks in the context of neuroimaging.

The results look interesting and indicate an improvement in sample efficiency based on the proposed subject-alignment procedure. This may be an interesting contribution to the workshop.

Typos:

?? in Section 2.5

**Reason For Not Giving Higher Score:**

N/A

**Reason For Not Giving Lower Score:**

The use of asynchronous inter-subject response alignment seems interesting and relevant to the workshop topic; the method appears to have potential.

**Reviewer Domain:**

machine learning

---

### Decision · Program_Chairs · 2024-03-02

Accept (Poster)